# The Improved Remediation Effect of the Combined Use of Earthworms with *Bacillus subtilis*-Loaded Biochar in Ameliorating Soda Saline–Alkali Soil

**DOI:** 10.3390/microorganisms13061243

**Published:** 2025-05-28

**Authors:** Zhichen Liu, Yingxin Huang, Qibiao Li, Luwen Zhang, Zhenke Liu, Zunhao Zhang, Yuxiang Chen

**Affiliations:** 1College of Biological and Agricultural Engineering, Jilin University, Changchun 130022, China; liuzhichen1996@163.com (Z.L.); luwen20@mails.jlu.edu.cn (L.Z.); liuzk23@mails.jlu.edu.cn (Z.L.); 2Northeast Institute of Geography and Agroecology, Chinese Academy of Sciences, Changchun 130102, China; huangyx@iga.ac.cn; 3Zhanjiang Experiment Station, Chinese Academy of Tropical Agricultural Science, Zhanjiang 524000, China; li-qibiao@catas.cn; 4The Electron Microscopy Center, Jilin University, Changchun 130012, China; zhangzunhao@jlu.edu.cn

**Keywords:** soda saline–alkali soil, *Bacillus subtilis*-loaded biochar, soil carbon fractions, key soil enzymes, soil microbial community

## Abstract

High pH, ${\mathrm{Na}}^{+}$, and $({\mathrm{CO}}_{3}^{2-}+{\mathrm{HCO}}_{3}^{-}$) are the primary characteristics of soda saline–alkali soil. Current strategies for ameliorating soda saline–alkali soil often involve the combined use of cow manure and maize straw, the addition of biochar (BC), and the inoculation of *Bacillus subtilis* (BS). In this study, *B. subtilis*-loaded biochar (BSC) was prepared using an adsorption technique. An incubation experiment was conducted. The treatments were as follows: soda saline–alkali soil amended with maize straw and cow manure (T_1_), which was used as a control; T_1_ supplemented with earthworms (T_2_); and T_2_ supplemented with BS (T_3_), BC (T_4_), or BSC (T_5_). After a 60-day incubation, T_5_ showed the most significant reduction in pH, ESP, and (${HCO}_{3}^{-}\;$ + $\;{CO}_{3}^{2-}$) concentrations, with reductions of 0.24 units, 3.26%, and 120 mg kg^−1^, respectively, compared to the T_1_ treatment. The content of soil humic acid, available potassium, and available nitrogen and the activities of β-glucosidase and urease were highest in T_5_, increasing by 33.5%, 70.1%, 26.1%, 19.0%, and 17.9%, respectively. Microbial sequencing analysis revealed that the *Bacillus* abundance in T_3_ was highest during the first 45 days (2.51–3.65%), while the *Bacillus* abundance in T_5_ peaked at 3.22% after the 60-day incubation. The soil that was cultivated for 60 days in the experiments was then used for planting alfalfa. T_5_ showed the highest alfalfa aboveground biomass and peroxidase, increasing by 30.1% and 73.1%, respectively, compared with T_1_. This study demonstrated that loading onto biochar is beneficial for the survival of *B. subtilis* in soda saline–alkali soil. When traditional organic materials are used, the combination of earthworms and *B*. *subtilis*-loaded biochar significantly alleviates the constraints of soda saline–alkali soil.

## 1. Introduction

Soil salinization results from both natural and anthropogenic factors and poses a significant threat to global food security and ecosystem stability [1]. Soda saline–alkali soil represents a predominant category of saline–alkaline soils, with the Songnen Plain in northeastern China constituting one of the three major global distribution areas for this specific soil type [2]. This type of soil is characterized by an elevated pH, exchangeable sodium percentage (ESP), and $({\mathrm{CO}}_{3}^{2-}+{\mathrm{HCO}}_{3}^{-}$) concentration, which exacerbates its adverse impacts on agricultural productivity and environmental health. The excessive uptake of Na^+^ by crops disrupts metabolic processes and reduces photosynthetic efficiency [3]. High salinity and alkalinity restrict nutrient use efficiency, which limits crop growth and reduces grain yield [4]. Given the vast areas of soda saline–alkali soil, the effective management and amelioration of these soils are crucial for food security.

The application of maize straw and cow manure as organic amendments is a common method for ameliorating saline–alkali soils [5]. However, the Songnen Plain is characterized by low temperatures, soil moisture, high salinity, and limited microbial activity [6], which hinder the decomposition and mineralization of maize straw and manure, restricting the formation of bioavailable carbon and nutrients [7,8]. The introduction of earthworms to low-salinity soils can accelerate organic matter decomposition [9]. For example, Wu et al. [10] conducted a field experiment and found that adding earthworms to saline–alkali soil with a pH of 7.39 and an EC of 744 μm cm^−1^ improved the soil’s physical and chemical properties and significantly increased the crop yield. Similarly, Zhang et al. [11] reported that earthworms reduced the pH and EC in soil with initial values of 8.33 and 645 μm cm^−1^, respectively, while increasing the available phosphorus (AP), soil organic carbon (SOC), and catalase (CAT) activity. Earthworm activity not only decreases soil salinity but also enhances the organic matter content, improving both the physical–chemical and biological properties of the soil. However, information is limited about the combined effects of earthworms, maize straw, and cow manure on soil carbon fractions and on the mitigation of salinity–alkalinity [12,13].

*B. subtilis* belongs to the plant growth-promoting rhizobacteria (PGPR), a group of beneficial bacteria known for their multifunctional roles, including nitrogen fixation, phosphate solubilization, the secretion of indoleacetic acid (IAA), the production of siderophores, and cytokinin synthesis. PGPR enhances agricultural productivity by improving the bioavailability of soil mineral nutrients and the secretion of plant growth hormones, such as IAA and cytokinin, which directly stimulate plant growth [14,15,16]. *Bacillus* spp. are the most widely applied growth-promoting bacteria in agriculture [17]. Studies indicate that *B. subtilis* is highly effective in remediating saline–alkali soil. Fermentation products, including ionized glutamic acid, significantly lower the soil pH, reduce salt crystallization, activate phosphorus, and increase the abundance of beneficial microorganisms [13,18,19]. Additionally, *Bacillus* reduces salt–alkali stress, enhances stress tolerance, increases the photosynthetic capacity, regulates metabolic pathways, and promotes plant growth [19,20]. However, exogenous growth-promoting bacteria often face environmental stress and intense competition with native microbes, leading to low survival rates and limited longevity [21,22]. Therefore, selecting a biologically friendly and sustainable carrier to enhance and prolong bacterial activity is crucial.

The investigation of environmentally friendly methods and materials for soil remediation has become a popular research topic. For example, Bi et al. [23] used *Bacillus halophilus* BH-8 combined with coal gangue to improve saline–alkali land, and Yu et al. [24] used biochar loaded with *Acinetobacter* to remediate petroleum-contaminated soil. Biochar, a porous, carbonaceous material, possesses a strong adsorption capacity and chemical stability, providing a good habitat for microorganisms. Additionally, it can be efficiently loaded with microorganisms and serve as an effective amendment for degraded and polluted soils [25,26]. Biochar reduces the pH, salinity, and Na^+^ content while increasing the cation exchange capacity (CEC), enzyme activity, nutrient availability, and microbial diversity [12,27,28]. Under saline–alkali stress, the application of biochar has been demonstrated to significantly improve crop growth (e.g., in tomatoes, eggplants, and corn) and enhance the yield and product quality [29,30]. Emerging evidence indicates that microbial communities immobilized on biochar present an increased soil colonization capacity, prolonged viability, and improved metabolic activity, thereby significantly contributing to soil remediation processes [31,32]. However, a comprehensive understanding of the synergistic effects of biochar-mediated microbial inoculation and its impacts on both saline–alkali soil amelioration and plant productivity is lacking in the scientific literature. This knowledge gap represents a crucial research frontier in the fields of soil bioremediation and sustainable agriculture.

In this study, *B. subtilis*-loaded biochar (BSC) was prepared and combined with earthworms, maize straw, and cow manure to remediate soda saline–alkali soil. An incubation experiment was carried out, and the incubated soil was used to grow yellow-flowered alfalfa (*Medicago falcata* L.). The aims of this study were to (1) investigate whether loading onto biochar is beneficial for the survival of *B. subtilis* in soda saline–alkali soil amended with traditional organic materials; (2) evaluate the improved remediation effects of the combination of earthworms and *B. subtilis*-loaded biochar on soil fertility enhancement, carbon pool regulation, saline–alkali stress mitigation, and microbial community reshaping; and (3) assess improvements in the biomass and key antioxidant enzyme activities of alfalfa. This study proposes a novel approach for ameliorating soda saline–alkali soil and provides valuable insights into the remediation mechanisms of the combined use of earthworms and BSC.

## 2. Materials and Methods

### 2.1. Materials

This experiment utilized *B. subtilis*, earthworms (*Eisenia fetida*), biochar prepared from maize straw (BC), air-dried maize straw, and cow manure to ameliorate soda saline–alkali soil. The *B. subtilis* strain used in this study was separated from the rhizosphere soil of silver-haired tree roots on Yongxing Island in the South China Sea. The soda saline–alkali soil used in this study was collected from Changling County, Jilin Province, China (44°45′ N, 123°45′ E), and has the following basic physicochemical properties: pH 9.68, organic matter 9.23 g kg^−1^, total nitrogen 0.87 g kg^−1^, available phosphorus 18.2 mg kg^−1^, and available potassium 166 mg kg^−1^. The combined soil samples were collected via a standard gouge auger, which zigzagged around the experimental area. This soil is considered a typical example of soda saline–alkali soil due to natural conditions and excessive human exploitation. The soil type is classified as solonetz based on the World Reference for Soil Resources. Air-dried cow manure and maize straw were collected from local farms; their physicochemical properties are described by Chen et al. [33]. The chemical characteristics of the maize straw and cow manure mixed substrate are as follows: pH 7.00; total organic carbon (TOC) 401 g kg^−1^ dry matter; total nitrogen (TN) 11.2 g kg^−1^ dry matter. Yellow-flowered alfalfa seeds were provided by the Northeast Institute of Geography and Agroecology, the Chinese Academy of Sciences. The biochar used in this study was purchased from Liyuan Environmental Protection (Beijing) Co., Ltd., Beijing, China. The experimental flowchart is shown in Figure 1.

### 2.2. Preparation of B. subtilis-Loaded Biochar, and Salt–Alkali Tolerance of B. subtilis

A phylogenetic tree of *B*. *subtilis* based on the 16S rDNA analysis system is shown in Appendix A. A single colony of *B. subtilis* was selected from the LB agar medium (The reagents used in this experiment are all conventional laboratory reagents, with analytical purity) and activated by incubation at 37 °C and 180 rpm for 6 h. The culture was then transferred at a 1:100 ratio into a fresh LB medium and incubated under the same conditions for 16 h. After centrifugation at 4000 rpm for 5 min, the supernatant was discarded, and the *B. subtilis* pellet was resuspended in distilled water to its original volume for further use (*B. subtilis* suspension, BS). The suspension was diluted in a series from 10–10^7^ using the spread plate method, yielding a final concentration of 2.7 × 10^9^ CFU mL^−1^. Salt and alkali tolerance tests for the strain were subsequently conducted. For the alkali tolerance test, 1% (*v*/*v*) of the activated bacterial culture were inoculated into LB media with pH values of 7.0, 7.5, 8.0, 8.5, 9.0, 9.5, and 10.0 and incubated at 37 °C for 48 h. Samples were collected at 4, 8, 12, 24, and 48 h, and their optical density (OD) at 600 nm was measured to assess bacterial growth under different pH conditions. For the salt tolerance test, the same inoculation procedure was followed using LB media containing NaCl at concentrations of 1%, 2%, 4%, 6%, and 8%, with samples collected and measured similarly. The biochar was mixed with the bacterial suspension at a mass-to-volume ratio of 1:5 and incubated for 8 h. The mixture was then filtered and dried to obtain BSC. Triplicates of 0.10 g of dried biochar were mixed with 5 mL of LB medium, followed by 10 min of ultrasonic treatment and 6 min of vortexing. The mixture was then spread on LB agar plates, and the colony-forming units were counted. The results indicated that the BSC concentration was 3.4 × 10^9^ CFU g^−1^, indicating high and uniform loading. Scanning electron microscopy with energy-dispersive spectroscopy (SEM-EDS) (Hitachi Regulus 8100, Hitachi, Tokyo, Janpan) was used to observe the surface morphology of the biochar. An elemental analysis was conducted using an elemental analyzer (Vario EL cube, Elementar, Hanover, Germany) to determine the contents of C, H, N, O, and S in the samples.

### 2.3. Experimental Design, Materials, and Sample Collection

The collected soil samples were air-dried, ground, and passed through a 2 mm sieve for further use. Earthworms were purchased from a local market, and mature earthworms weighing 0.3 to 0.5 g were selected for the experiment. The soil incubation experiment lasted 60 days, with samples taken every 15 days. After 60 days of soil incubation, the soil was used to grow alfalfa, with a growth period of 30 days. Each treatment was performed in triplicate. The samples were stored at −80 °C prior to laboratory testing. The treatments were as follows: T_1_, 30 g straw and 20 g manure per kilogram of soil, used as a control; T_2_, 6 earthworms per kilogram of soil based on T_1_; T_3_, 25 mL of BS suspension per kilogram of soil based on T_2_; T_4_, 10 g BC per kilogram of soil based on T_2_; and T_5_, 10 g BSC per kilogram of soil based on T_2_. After 60 days of incubation, the amended soil was transferred to polyethylene pots, each containing 2500 g of soil. Alfalfa seeds were selected and sun-dried for 1 h, followed by soaking in 10% NaClO solution for 30 min. The seeds were rinsed three times with tap water and distilled water and then soaked in distilled water overnight. Twenty seeds were sown in each pot and covered with a thin layer of soil. Seven days after germination, thinning was performed to ensure ten uniform seedlings per pot. Samples were taken for testing after 30 days of alfalfa growth. The samples were stored at −80 °C prior to laboratory testing. The plants were grown in a greenhouse with temperatures maintained at 18–25 °C, an average daily light duration of approximately 6.5 h, a relative air humidity of approximately 35%, and a soil moisture of approximately 10%.

### 2.4. Physical and Chemical Analyses

The incubated soil samples were suspended in pure water at ratios of 1:5 and 1:2.5 for the determination of electrical conductivity (EC) and pH, respectively. The suspensions were shaken for 10 min, and the EC and pH values were measured using conductivity and potentiometric methods, respectively. The cation exchange capacity (CEC) and exchangeable sodium (ENa) were determined using sodium acetate flame photometry [34]. The concentration of $({\mathrm{CO}}_{3}^{2-}+{\mathrm{HCO}}_{3}^{-}$) was measured following the method described by Wang et al. [35]. The exchangeable sodium percentage (ESP) refers to the percentage of exchangeable sodium ions in the total exchangeable cations and is calculated as(1)ESP=100% × ENa/CEC

The activities of β-glucosidase, sucrase, and urease were measured using 4-nitrophenyl-β-d-glucopyranoside, sucrose, and urea as substrates, following the methods of Cai et al. [36] and Kaur et al. [37]. Under measurement conditions, sucrase activity is expressed as mg glucose g^−1^ (24 h)^−1^, β-glucosidase activity as μg p-nitrophenol g^−1^ h^−1^, and urease activity as mg ${NH}_{4}^{+}$ -N g^−1^ (24 h)^−1^. The available nitrogen (AN), available phosphorus (AP), and available potassium (AK) in the soil were measured following the method described by Wan et al. [38]. The sodium content in the plant tissues was determined according to Ramandi et al. [39]. The activities of catalase (CAT), superoxide dismutase (SOD), and peroxidase (POD) were measured based on the method of Liu et al. [40], with enzyme activity defined as a change in absorbance of 0.01 per gram of sample per milliliter of the reaction system. Soil carbon fractions were determined following the method described by Ai et al. [41].

### 2.5. Microbial Community Structure Analysis

DNA extraction and purification were performed following the method described by Chen et al. [33]. Specific primers with barcodes were used for PCR amplification. The primers 338F (5′-ACTCCTACGGGAGGCAGCAG-3′) and 806R (5′-GGACTACHVGGGTWTCTAAT-3′) were used to amplify the V3–V4 region of the bacterial 16S rRNA gene, while ITS1F (5′-CTTGTCATTTAGAGAAGTAA-3′) and ITS2R (5′-GTGCGTTCTTCATCGATGC-3′) were used to amplify the ITS1 region of the fungal gene. The PCR products were recovered using a 2% agarose gel and purified with the AxyPrep DNA Gel Extraction Kit (Axygen Biosciences, Union City, CA, USA). The purified products were quantified using the Quantus™ Fluorometer (Promega, Madison, WI, USA). Sequencing was performed using the Sequencing equipment (Illumina MiSeq PE300, Illumina, San Diego, CA, USA). The FUNGuild (Version 1.0) tool was used to classify fungal trophic modes and functional groups [42]. The FAPROTAX database was used to predict the ecological functions of soil bacteria [43].

### 2.6. Statistical Analysis

The results for the soil carbon fractions, enzyme activities, available nutrients, salinity–alkalinity indices, microbial alpha diversity indices, and alfalfa-related indicators are expressed as the mean ± standard deviation (n = 3). Multiple comparisons were performed using Tukey’s HSD test to determine significant differences among groups. Different lowercase letters (e.g., a, b, c, d) indicate statistically significant differences (*p* < 0.05). The data on soil bacteria and fungi were processed using the Majorbio Cloud Platform (https://cloud.majorbio.com). The alpha diversity indices (Chao 1, Shannon, and Ace) were calculated using the mothur software (https://www.mothur.org/wiki/Calculators) accessed on 4 March 2025, following the method of Partrick et al. [44]. The LEfSe (Linear discriminant analysis Effect Size) (http://huttenhower.sph.harvard.edu/LEfSe) accessed on 15 March 2025, was used to identify microbial taxa with significant differences in abundance between groups, from phylum to genus levels, with LDA > 3.5 and *p* < 0.05. The correlation analysis picture was created through https://www.chiplot.online/, accessed on 26 March 2025.

## 3. Results

### 3.1. Growth Status of Free B. subtilis and Characterization of BSC

*B. subtilis* exhibited varying activity levels in LB media across pH values ranging from 7 to 9.5 (Figure 2a). During the 0–48 h period, its activity initially increased, followed by a decrease, with higher pH levels resulting in reduced activity. Optimal activity was observed at pH 7, whereas at pH 10, activity was nearly undetectable. Activity decreased as the NaCl concentration increased (Figure 2b). At 8% NaCl, the activity was minimal and nearly undetectable. At NaCl concentrations of 1–2%, the activity initially increased but later decreased, whereas at 4–6%, the activity consistently increased over time. The surface morphologies of BC and BSC were analyzed using SEM-EDS (Figure 2c,d). Uniform *B. subtilis* cells adhered to the rod walls of BSC. An elemental analysis revealed that, compared to BC, BSC had higher sulfur (0.14% to 0.22%), carbon (70.4% to 70.9%), and nitrogen (0.85% to 1.28%) contents (Figure 2e,f).

### 3.2. Soil Carbon Fractions

The initial soil organic carbon (SOC) content was 5.29 g kg^−1^ (Appendix A). Compared to the T_1_ treatment, the T_4_ and T_5_ treatments significantly (*p* < 0.05) increased the SOC content at all the time points (Figure 3a). The SOC content decreased over time in all the treatments, with the largest decrease observed in T_5_, which decreased by 3.49 g kg^−1^ from day 15 to day 60 during the incubation period. The initial dissolved organic carbon (DOC) content was 178.3 mg kg^−1^ (Appendix A). The DOC content across the treatments followed the order T_3_ > T_1_ > T_2_ > T_4_ > T_5_ (Figure 3b), with significant differences (*p* < 0.05) between treatments. The DOC content showed a decreasing trend over time for all treatments, with the greatest reduction observed in T_4_, which decreased by 70.4 mg kg^−1^ from day 15 to day 60 of the incubation period. The initial particulate organic carbon (POC) content was 0.37 g kg^−1^ (Appendix A). During the first 45 days, the POC levels in the T_4_ and T_5_ treatments were significantly higher than those in the T_1_, T_2_, and T_3_ treatments (*p* < 0.05), with no significant differences among the T_1_, T_2_, and T_3_ treatments (Figure 3c). After 60 days of incubation, the POC content in the T_1_ treatment was significantly (*p* < 0.05) higher than those in the T_2_ and T_3_ treatments. The initial readily oxidizable organic carbon (ROC) content was 2.32 g kg^−1^ (Appendix A). The ROC content increased over time for all treatments, with the highest ROC content observed in T_4_ (8.30 g kg^−1^) and the lowest in T_3_ (7.12 g kg^−1^) at the end of incubation (Figure 3d). No significant differences in the ROC content were found between the T_1_, T_2_, and T_3_ treatments at any time point, whereas the ROC content in the T_4_ treatment was significantly (*p* < 0.05) higher than the T_1_, T_2,_ and T_3_ treatments. The initial fulvic acid (FA) content was 0.34 g kg^−1^ (Appendix A). During the first 30 days, no significant differences were observed among the T_1_, T_2_, T_3_, T_4_, and T_5_ treatments (Figure 3e). However, after 45 days of incubation, the FA contents in the T_2_ treatment differed significantly (*p* < 0.05) from those in the T_4_ and T_5_ treatments, although there were no significant differences between the T_4_ and T_5_ treatments. At 60 days, the FA content in the T_1_, T_2,_ and T_3_ treatments was significantly (*p* < 0.05) higher than that in the T_4_ and T_5_ treatments. The initial humic acid (HA) content was 1.35 g kg^−1^ (Appendix A) and tended to increase over time among all the treatments (Figure 3f). After 60 days of incubation, the HA content in the T_4_ and T_5_ treatments was significantly (*p* < 0.05) higher than that in the T_1,_ T_2,_ and T_3_ treatments. The greatest increase was observed in the T_5_ treatment, which increased by 1.05 g kg^−1^ from day 15 to day 60 and increased 33.51% in comparison to T_1_.

### 3.3. Key Soil Enzyme Activities

The initial β-glucosidase activity in the soil was 17.6 μg p-nitrophenol g^−1^ h^−1^ (Appendix A). The β-glucosidase activity decreased over time in all the treatments (Figure 4a). The T_4_ and T_5_ treatments exhibited the highest β-glucosidase activity from day 15 to day 60, the T_1_ treatment showed the lowest activity, and the T_2_ and T_3_ treatments were higher than the T_1_ treatment. After 60 days of incubation, the activity of β-glucosidase in T_5_ was highest, increasing by 19.1% in comparison to T_1_. The initial invertase activity was 0.93 mg glucose g^−1^ (24 h)^−1^ (Appendix A). The activity of soil invertase decreased over time in all the treatments (Figure 4b). Across all the time points, the highest invertase activity was observed in the T_4_ treatment, while the lowest activity was in the T_3_ treatment. Compared with the T_1_ treatment, the T_2_ treatment consistently resulted in greater invertase activity. After 60 days of incubation, the T_4_ treatment exhibited the highest invertase activity (10.6 mg glucose g^−1^ (24 h)^−1^), whereas the T_3_ treatment showed the lowest invertase activity (9.08 mg glucose g^−1^ (24 h)^−1^). The initial urease activity in the soil was 0.19 mg $NH_{4}^{+}$ -N g^−1^ (24 h)^−1^ (Appendix A). The urease activity tended to increase over time in all the treatments (Figure 4c). Urease activity was ranked at all the time points as T_5_ > T_4_ > T_3_ > T_2_ > T_1_. By the end of the 60-day incubation, the urease activity in the T_3_, T_4_, and T_5_ treatments was significantly (*p* < 0.05) higher than that in the T_1_ treatment. After 60 days of incubation, the activity of urease in T_5_ was highest, increasing by 17.9% in comparison to T_1_.

### 3.4. Available Nutrients and Elimination of Saline–Alkali Barriers

The initial available potassium (AK) content was 150 mg kg^−1^ (Appendix A). Significant differences (*p* < 0.05) in the AK content were observed in the T_1_, T_3_, T_4_, and T_5_ treatments, with the following ranking: T_5_ > T_4_ > T_3_ > T_2_ > T_1_ (Figure 3d). The T_5_ treatment exhibited the highest AK content, reaching 478 mg kg^−1^, increasing by 70.1% in comparison to T_1_. The initial available phosphorus (AP) content was 18.5 mg kg^−1^ (Appendix A). The highest AP content was found in the T_2_ treatment (50.0 mg kg^−1^) (Figure 4e). The initial available nitrogen (AN) content in the soil was 33.0 mg kg^−1^ (Appendix A). Compared with the T_1_ treatment, the T_4_ and T_5_ treatments significantly (*p* < 0.05) increased the AN content_,_ following the order of T_5_ > T_4_ > T_3_ > T_2_ > T_1_ (Figure 4f). The T_5_ treatment had the greatest AN content at 61.0 mg kg^−1^, which was significantly (*p* < 0.05) higher than the other treatments and an increase of 26.1% in comparison to T_1_. The T_2_ treatment significantly (*p* < 0.05) reduced the soil EC $({\mathrm{CO}}_{3}^{2-}+{\mathrm{HCO}}_{3}^{-}$) and the exchangeable sodium (ENa) (Table 1) in comparison to the T_1_ treatment, whereas the T_3_, T_4_, and T_5_ treatments reduced the soil pH, electrical conductivity (EC), and $({\mathrm{CO}}_{3}^{2-}+{\mathrm{HCO}}_{3}^{-}$) concentration in comparison to the T_2_ treatment. The T_5_ treatment resulted in the greatest decreases in pH, ESP, and $({\mathrm{CO}}_{3}^{2-}+{\mathrm{HCO}}_{3}^{-}$) concentration, with reductions of 0.24 units, 3.26%, and 120 mg kg^−1^, respectively, in comparison to T_1_. The T_4_ and T_5_ treatments exhibited the greatest increases in the cation exchange capacity (CEC), with increases of 19.9% and 24.5%, respectively.

### 3.5. Microbial Composition and Analysis of Differences

Using the Ace index as an example, differences in alpha diversity were observed among the treatments (Table 2). The addition of earthworms increased the soil bacterial diversity at any time point, but the soil alpha diversity decreased in the T_3_ treatment in comparison to the T_2_ treatment. However, adding BC effectively enhanced the microbial diversity. The combination of earthworms, BS, and BC adjusted the soil bacterial community structure, with earthworms and BC likely increasing the relative abundance of beneficial bacteria. During the early stages of incubation, the bacterial diversity in T_5_ was relatively low but gradually increased over time. The Ace index was significantly lower in the T_4_ and T_5_ treatments than in the T_2_ treatment. On day 30, the differences between the treatments were not significant; by day 45 to 60, the Ace index of the T_4_ and T_5_ treatments was higher than that of T_1_. The Ace of T_3_ was lower than that of the T_2_ treatment at all the time points.

Soil samples were taken at four time points throughout the incubation period for the sequencing analysis. The relative abundances of the bacterial and fungal communities are shown in Figure 4. At the phylum level, the 10 most abundant bacteria, which included Actinobacteriota, Proteobacteria, Chloroflexi, Firmicutes, Bacteroidota, Acidobacteriota, Gemmatimonadota, and Myxococcota, accounted for 90.6–95.3% of the total bacterial abundance (Figure 5a). On day 15, the relative abundance of Actinobacteriota was highest in the T_3_ treatment, while Firmicutes had the highest abundance in the T_5_ treatment. The average relative abundance of Firmicutes followed the order T_5_ > T_3_ > T_2_ > T_4_ > T_1_. Among the fungal communities, Ascomycota, Basidiomycota, and unclassified_k_Fungi constituted the majority, accounting for 96.9–97.4% of the total fungal abundance (Figure 5b). Across all the treatments, Basidiomycota had the highest relative abundance, peaking at 15.4% on day 15.

At the genus level, the 15 most abundant bacterial genera included *Pontibacter*, *norank_f_JG30-KF-CM45*, *norank_f_A4b*, *norank_f_norank_o_Vicinamibacterales*, *Skermanella*, *norank_f_Anaerolineaceae*, *Lysobacter*, and *Bacillus*, accounting for 19.6–29.3% of the total bacterial abundance (Figure 5c). The relative abundance of *Bacillus* peaked in the T_3_ treatment after 45 days (3.65%) and steadily increased in the T_5_ treatment throughout the incubation period, reaching its highest level (3.22%) on day 60. This suggested that BS produced short-terms effects when introduced, whereas BSC maintained a higher relative abundance of Bacillus and prolonged its survival time. The relative abundance of *Streptomyces* was greater in the T_3_ treatment throughout the incubation period and in the T_5_ treatment from day 30 to 60. A Venn diagram (Figure 6d) showed that all the treatments shared 2941 OTUs, accounting for 21.38% of the total amount. T_4_ had the greatest number of unique species (1432 OTUs, 10.41%). The LefSe analysis of bacterial communities after 60 days of incubation (Figure 6a) revealed that the T_2_ treatment influenced Desulfobacterota, whereas the T_5_ treatment affected Firmicutes, Bacillales, and Bacilli. In contrast, the T_3_ treatment impacted Actinomarinates and Gammaproteobacteria. Although the earthworm, BS, and BSC combinations altered the bacterial community structures at the phylum level, the changes in fungal communities were less pronounced. At the genus level (Figure 6c), *Bacillus*, *Solibacillus*, and *Romboutsia* significantly differed in the T_5_ treatment compared to other treatments (*p* < 0.05), while Pseudomonas significantly differed in the T_3_ treatment (*p* < 0.05).

For fungal communities, the most abundant genera included *Aspergillus*, *Tausonia*, *Scopulariopsis*, *unclassified_k_Fungi*, *Mycothermus*, *unclassified_c_Sordariomycetes*, *Kernia*, *unclassified_o_Sordariales*, and *Thelebolus*, accounting for 40.3–71.1% of the total fungal abundance (Figure 5d). Compared to the T_1_ treatment, the T_2_ treatment increased the relative abundance of *Mycothermus* at all the time points. After 60 days of incubation, the relative abundance of *Mycothermus* was highest in T_5_. During the first 45 days, the T_2_ treatment had a higher *Thelebolus* abundance than the T_1_ treatment, with the highest abundance consistently found in the T_4_ treatment. The significance testing of fungal genera (Figure 6e) showed significant differences in *Aspergillus* between the T_2_ and T_5_ treatments and in Zopfiella between the T_2_ and T_4_ treatments (*p* < 0.05). A Venn diagram (Figure 6f) showed that all the treatments shared 316 OTUs, accounting for 19.2% of the total, with the highest number of unique species (209 OTUs, 12.7%) found in the T_4_ treatment.

### 3.6. Microbial Composition Prediction and Correlation Analysis of Microbial Community Function

Using the FAPROTAX model, the functional prediction of the soil bacterial microbiota (Figure 7a) showed that the T_2_ treatment resulted in greater ureolysis activity than the T_1_ treatment at all the time points. On day 15, ureolysis in the T_3_ treatment was significantly (*p* < 0.05) higher than in the T_2_ treatment, and by day 60, the T_5_ treatment showed the highest ureolysis activity. The T_2_, T_3_, and T_5_ treatments exhibited higher nitrate-reduction activity than the T_1_ treatment, while the T_3_ and T_5_ treatments outperformed the T_2_ treatment. For the degradation of aromatic hydrocarbons, aliphatic compounds, and hydrocarbons, the application of earthworms, BS, and BSC effectively reduced the negative impact of these compounds. Using the FUNGuild model (Figure 7b), the functional prediction of fungal communities revealed that the T_5_ treatment had the highest relative abundance of saprotrophs on days 30 and 60, and the T_3_ treatment had the highest relative abundance on day 45. The T_2_ treatment exhibited higher dung saprotroph abundance and lower animal pathogen abundance in comparison to the T_1_ treatment.

The correlation analysis of the microbial genera and soil properties (Figure 7c) revealed strong positive correlations among *Bacillus*, *Mycothermus*, *Vicinamibacterales*, and *Aspergillus*. AN was positively correlated with *Mycothermus* and *Aspergillus*, whereas AP was correlated with *Mycothermus*, *Aspergillus*, and *A4b*. AK was positively correlated with *Bacillus*, and pH was positively correlated with *Scopulariopsis*, *Lysobacter*, and *Pontibacter*. ENa was positively correlated with *Scopulariopsis*, *Tausonia*, and *Pontibacter* (*p* < 0.05, r > 0.03).

### 3.7. Growth and Resistance of Alfalfa

The morphology of the different alfalfa treatments after 30 days of growth is shown in Figure 8a. Compared to the T_1_ treatment, the T_3_, T_4_, and T_5_ treatments significantly (*p* < 0.05) increased the aboveground and underground biomass of alfalfa (Figure 8b), with the order of improvement degree following T_5_ > T_4_ > T_3_ > T_2_ > T_1_. Compared to the T_1_ treatment, the T_5_ treatment had the greatest improvement on aboveground and underground biomass, reaching 247 mg plant^−1^ and 36.2 mg plant^−1^, with an increase of 30.2% and 34.1%, respectively. Compared to the T_1_ treatment, T_2_, T_3_, T_4_, and T_5_ significantly (*p* < 0.05) increased the plant height of alfalfa, following the order of T_5_ > T_4_ > T_3_ > T_2_ > T_1_. Significant (*p* < 0.05) differences in plant height were observed between all the treatments, but no significant differences were observed in root length. The T_5_ treatment resulted in the greatest plant height, reaching 9.99 cm, whereas the T_2_ treatment resulted in the greatest root length, reaching 3.09 cm (Figure 8c). Compared to the T_1_ treatment, the T_3_, T_4_, and T_5_ treatments significantly (*p* < 0.05) reduced the sodium content in both the underground and aboveground portions of the alfalfa (Figure 8d). The aboveground alfalfa of the T_5_ treatment showed the greatest decline in the sodium content, with a decrease of 16.8% in comparison to T_2_. Compared to the T_1_ treatment, the T_3_ and T_5_ treatments significantly increased (*p* < 0.05) the POD activity in the aboveground parts of the alfalfa (Figure 8e), with increases of 56.2% and 73.1%, respectively. The POD activity in the T_5_ treatment was significantly greater than that in the T_1_, T_2_, and T_4_ treatments. Compared to the T_1_ treatment, the T_4_ and T_5_ treatments significantly increased (*p* < 0.05) the SOD activity in the aboveground part of the alfalfa (Figure 8f), reaching 1381 and 1491 U g^−1^, respectively. Compared to the T_1_ treatment, the T_2_, T_3_, T_4_, and T_5_ treatments significantly (*p* < 0.05) increased the CAT activity in the aboveground part of the alfalfa. T_3_ and T_5_ presented the greatest increase, at 163 and 181 U g^−1^, respectively.

## 4. Discussion

### 4.1. Analysis of Salt Alkali Tolerance of Free B. subtilis and Characterization of BSC

The salt and alkali tolerance tests for free-living *B. subtilis* showed that it was able to survive at environmental pH levels ranging from 7 to 9.5, with the highest activity observed at pH 7. It also survived NaCl concentrations of 1–6%, with activity decreasing as the NaCl concentration increased. The tubular structures of the biochar had diameters of approximately 5–10 μm, much larger than the size of the *B. subtilis* cells. Its porous structure and surface wrinkles provided a favorable environment for bacterial attachment and colonization. Sulfur- and nitrogen-containing compounds are known secondary metabolites of *B. subtilis* [45], and the elemental differences between BSC and BC further confirmed that *B. subtilis* was successfully loaded onto the biochar. Multiple samples of BSC were plated on LB agar to determine bacterial colony counts, indicating that the BSC concentration was 3.4 × 10^9^ CFU g^−1^. The loading of *B. subtilis* onto the biochar was efficient and uniform.

### 4.2. Analysis of Dynamic Changes in Soil Carbon Fractions

The maize straw and cow manure amendments supplied the soil with ample organic carbon. After 60 days of incubation, the earthworm—BSC treatment exhibited the highest degradation rates of organic carbon from straw and manure. DOC, a labile form of organic carbon, is readily utilized by soil microbes, providing essential nutrients and playing a crucial role in nutrient cycling [41]. The results indicated that straw and manure initially supplied ample DOC, but the introduction of earthworms reduced the DOC level. The application of exogenous BS further reduced the ability of the original microorganisms to utilize DOC. T_4_ resulted in the greatest reduction in DOC, suggesting that biochar enhanced the microbial DOC utilization efficiency. Introducing BSC and earthworms retained more DOC than BC and earthworms, preserving more available carbon for both soil microbes and plants.

POC is essential for maintaining soil structure and water-holding capacity [46]. The application of BC and BSC improved the POC level, whereas the addition of earthworms and BS led to a decrease in the POC content during the later incubation stages. Both BC and BSC enhanced the soil stability, with no significant differences between them. ROC, a highly reactive form of organic carbon, reflects early soil condition changes, serves as a primary nutrient source for plants, and plays a key role in stabilizing the soil organic carbon pool [46]. The results showed that the application of earthworms combined with BS did not significantly affect the soil organic carbon oxidation level, whereas the application of earthworms combined with BC and BSC increased the amount of plant-available organic carbon.

FA is crucial for promoting mineral decomposition and nutrient release. It has strong ion exchange and complexing abilities, enhancing soil water retention and nutrient availability. FA can transform into HA through humification [47]. HA contributes to soil fertility and plant growth, has high stability, and is a key component of soil organic matter, playing an important role in stabilizing the soil carbon pool [48]. The application of BC increased the HA content while reducing FA formation. The application of earthworms combined with BSC accelerated SOC decomposition from maize straw and manure, increasing the POC content and promoting carbon conversion into HA, thus stabilizing the soil carbon pool. This finding indicated that earthworms combined with BSC accelerated the decomposition of straw and cow manure, stabilized the soil carbon pool, and provided more available carbon for soil microorganisms and plants.

### 4.3. Analysis of Key Soil Enzymes, Available Nutrients, and Elimination of Salt–Alkali Barriers

β-glucosidase, a part of the cellulase degradation system, plays a key role in cellulose breakdown [49]. The application of earthworms combined with BS, BC, and BSC accelerated cellulose degradation in maize straw and manure, supplying energy for microbial activities. Earthworms combined with BSC most effectively accelerated cellulose decomposition in mixed maize straw and cow manure, generating organic carbon accessible to microbes and plants. Invertase hydrolyzed the sucrose in the soil, producing glucose and fructose [50]. Earthworms combined with BC increased the monosaccharide conversion rate, whereas BS reduced it, suggesting that earthworms combined with BC enhanced the microbial energy supply and soil metabolic activity. Urease catalyzed urea hydrolysis, generating bioavailable ammonium [51]. The application of earthworms combined with BS, BC, and BSC increased ammonium conversion rates under conventional organic amendment conditions, with the BSC combination exhibiting the highest rate. The findings demonstrated that the application of earthworms combined with BSC can effectively enhance soil nitrogen availability.

Soil AN, AP, and AK are essential plant-available nutrients that support normal growth and development [52]. The individual addition of earthworms significantly increased the AP and AN; conversely, when the earthworms were combined with BS and BC, the AK and AN increased. The combined effect of earthworms and BSC exhibited synergy, which was consistent with findings of Wu et al. [53]. However, earthworms combined with BS, BC, and BSC had no significant effect on available phosphorus levels. The application of BC supplies energy for carbon and nitrogen enzyme activities in soil [54]. Earthworms combined with BSC produced more nitrogen and phosphorus than earthworms combined with BC, providing more nutrients for the growth of plants.

The addition of earthworms effectively reduced the soil pH and concentrations of $({\mathrm{CO}}_{3}^{2-}+{\mathrm{HCO}}_{3}^{-}$), ENa, and ESP, whereas the combined application of BS decreased the soil pH and concentrations of EC, and $({\mathrm{CO}}_{3}^{2-}+{\mathrm{HCO}}_{3}^{-}$). BC reduced the pH, $({\mathrm{CO}}_{3}^{2-}+{\mathrm{HCO}}_{3}^{-}$) concentration and ESP, but the EC and ENa increased. The combination of earthworms and BSC was the most effective at alleviating soil salinity and alkalinity. The application of earthworms combined with BS enriched soil beneficial cations, while the addition of earthworms and BC and BSC significantly increased soil CEC, thereby reducing ESP. Compared to application of BC, the combination of earthworms and BSC showed superior effects on mitigating soil salinity and alkalinity.

The addition of earthworms combined with BC and BSC promoted the formation of HA and FA. The adsorption of Na^+^ on the surface of humic colloids, which contain numerous negatively charged sites, reduced ENa [55,56]. Amini et al. [57] and Wu et al. [58] reported that the application of organic materials reduced the ENa content, which was associated with the formation of macro-aggregates. Earthworms improved flocculation between soil particles, facilitating the formation of macro-aggregates and reducing capillarity [59,60]. A greater proportion of macro-aggregates was attributed to the dominance of Ca^2+^ over Na^+^ at clay exchange sites [61]. This ionic exchange enhances soil aggregation by promoting stronger bonds between soil particles, improving the soil structure and stability. The findings might explain the lower CEC values observed in the T_5_ treatment and the reduced ESP value in the earthworm and BSC treatment.

As shown in Figure 9, SOC, ROC, HA, POC, AN, AK, β-glucosidase, and urease were negatively correlated with pH and ESP. SOC, HA, POC, AN, AK, β-glucosidase, and urease showed a highly significant negative correlation with pH (*p* < 0.01), while AK, β-glucosidase, and urease showed a significant negative correlation with ESP (*p* < 0.01). Therefore, among the indicators measured in this study, AK, β-glucosidase, and urease significantly reduced the main soil salinity indicators, thereby improving the soil quality.

### 4.4. The Analysis of the Diversity, Composition, and Functional Prediction of the Microbial Community

Bacterial diversity was inversely correlated with DOC (Figure 3b), suggesting that bacterial growth depleted DOC. SOC decomposition was likely driven by increased bacterial diversity (Figure 3a), while the reduction in soil salinity and alkalinity was associated with enhanced diversity (Table 1). Fungal α-diversity responded differently to the treatments across the time points. Compared with conventional organic improvement, the introduction of earthworms increased the alpha diversity of the soil bacterial and fungal communities. Earthworms combined with BC and BSC reduced the fungal α-diversity in the early stages, but the differences diminished by day 30. The T_5_ treatment resulted in the highest Ace value during the later incubation stages. In contrast, BS reduced the Ace value, likely due to the invasive effects of exogenous species [62]. However, loading BS onto BC mitigated its negative effects on microbial diversity, enabling the decomposition of straw and cow manure in soil. Although the Chao1, Shannon, and Ace indices varied slightly, the trends were overall consistent.

The relative abundance and function of the bacterial communities varied across treatments and time points. Firmicutes enhanced soil health [63], whereas Basidiomycota served as an indicator of effective land use for soil carbon accumulation and functionality [64]. The LefSe analysis of microbial communities after 60 days of incubation (Figure 6a) revealed that the T_2_ treatment influenced Desulfobacterota. The T_5_ treatment affected Firmicutes and significantly increased the relative abundance of Bacillus, *Solibacillus, and Romboutsia*. The results demonstrated that earthworms combined with BSC increased the relative abundance of these beneficial microorganisms, whereas the addition of earthworms alone did not result in corresponding changes in these microorganisms. The application of earthworms combined with BSC improved the soil microbial community structure more effectively and demonstrated that loading onto biochar was beneficial for the survival of *B. subtilis* in soda saline–alkali soil. Although the application of earthworms combined with BS and BSC altered the bacterial community structures at the phylum level, the changes in fungal communities were less pronounced. When the LDA value was set to 2.0, only *fungi_cls_Incertac_sedis* was affected by T_5_. *Streptomyces* provides organic nitrogen and phosphorus directly to plants [65]. The treatments with BS increased the relative abundance of *Streptomyces*, possibly contributing to increased soil nutrient availability (Figure 3d–f). *Gemmatimonadaceae*, known for its phosphate-solubilizing properties and crop resistance enhancement [66], presented a greater relative abundance in T_2_ than in T_1_ at all the time points, explaining the increase in available phosphorus in T_2_ (Figure 4d). This finding might also explain why the addition of earthworms increases the available phosphorus content in the soil. *Saccharopolyspora* produces insecticidal compounds, such as spinosad, which degrade rapidly in the environment and contribute to the breakdown of carbon and nitrogen. *Rhizobiaceae* demonstrates nitrification ability [67], with higher relative abundances found in the T_2_, T_3_, T_4_, and T_5_ treatments compared to the T_1_ treatment, which could explain the increased nitrification and available nitrogen (Figure 4d). Earthworms combined with BSC could increase the soil available nitrogen.

Fungal communities varied across the treatments and the time points. *Mycothermus* promotes plant growth by increasing the uptake of nitrogen, phosphorus, and potassium [68]. The relative abundance of *Mycothermus* in T_2_ was significantly greater (*p* < 0.05) than in T_1_. On day 30, the relative abundance of *Aspergillus* was highest in the T_3_ and T_5_ treatments, suggesting that earthworms combined with BS and BSC enhanced the activity of *Aspergillus*, a fungus involved in organic matter mineralization and humification [69]. This might explain the rapid decrease in SOC and the increase in HA formation in the T_5_ and T_3_ treatments in comparison to the T_1_ and T_2_ treatments (Figure 3a,f). *Thelebolus* produces hydrolytic and oxidative enzymes that decompose organic matter, participate in nutrient cycling, and influence soil physicochemical properties and enzyme activities [70]. Compared to the T_1_ treatment, the T_2_ treatment was more positively correlated with nitrogen respiration at all stages. The T_3_ and T_5_ treatments were more positively correlated with nitrogen respiration in comparison to the T_1_ treatment (Figure 7a), indicating that the addition of earthworms accelerated nitrogen decomposition; the addition of BS and BSC further enhanced this effect. The application of earthworms combined with BSC resulted in greater ureolysis activity during the later stages of incubation, which might explain the differences in urease activity (Figure 4d) and available nitrogen (Figure 4e) among the treatments. The results indicated that the combination of earthworms and BSC improved the microbial community structure of the soda saline–alkali soil, increased the abundance of beneficial microorganisms, and contributed to disease suppression and soil health.

The application of earthworms accelerated SOC decomposition, and the addition of BC further enhanced this process, likely due to increases in soil nutrients and enzyme activities (Figure 3a and Figure 4a–f). *Zopfiella*, an important soil health indicator, was more abundant in the T_2_, T_4_, and T_5_ treatments than in the T_1_ treatment, suggesting that the application of earthworms combined with BC and BSC could improve soil health conditions. The application of earthworms combined with BS and BSC altered the microbial community structures, increased the abundance of beneficial fungi and bacteria, and promoted organic matter decomposition (Figure 5d). These changes might increase soil enzyme activity and nutrient availability (Figure 4), alleviating soil salinity and alkalinity (Table 1).

### 4.5. Analysis of Alfalfa Growth

The combination of earthworms and BSC resulted in the greatest improvement in the aboveground and belowground biomass of alfalfa, with significant reductions in the Na content of both. POD, SOD, and CAT are key antioxidant enzymes, and an increase in activity indicates improved plant stress resistance [71]. The aboveground alfalfa POD, SOD, and CAT activities in the T_5_ treatment were significantly greater than those in the T_1_ and T_2_ treatments. The findings demonstrated that the application of earthworms alone did not significantly improve the aboveground alfalfa POD and SOD activities, whereas when earthworms were combined with BSC, the activities of all three antioxidant enzymes significantly increased. The application of earthworms combined with BS, BC, and BSC effectively alleviated sodium stress in alfalfa, likely through reduced soil ENa and increased CEC. The increase in alfalfa biomass and length in the aboveground and belowground parts might be attributed to increased soil nutrient levels, the partial mitigation of salinity and alkalinity, the increased abundance of resistance-related microbes, and reduced sodium stress. These factors collectively improved alfalfa resistance, leading to increased biomass. The application of earthworms combined with BC and BS had beneficial effects on alfalfa growth, but the three antioxidant enzymes exhibited varying activity trends across the T_1_, T_2_, T_3_, and T_4_ treatments. The combination of earthworms and BSC promoted optimal alfalfa growth, resulting in the greatest improvement in biomass.

## 5. Conclusions

The results of the present study demonstrated that the application of earthworms was beneficial to the improvement of soda saline–alkali soil, with the *B. subtilis*-loaded biochar most effectively alleviating soda saline–alkali stress. At the end of the incubation period, the T_5_ treatment showed the greatest reductions in pH, ESP, and $({\mathrm{CO}}_{3}^{2-}+{\mathrm{HCO}}_{3}^{-}$) content, decreasing by 0.24 units, 3.16%**,** and 120 mg kg^−1^, respectively. The content of soil humic acid, available potassium, available nitrogen, β-glucosidase activity, and urease activity were highest in the T_5_ treatment, increasing by 33.5%, 70.1%, 26.1%, 19.1%, and 17.89%, respectively. After 60 days of incubation, microbial sequencing revealed that the relative abundance of *Bacillus* peaked at 3.22% in the T_5_ treatment. The application of BSC maintained a high relative abundance of *Bacillus* and extended its survival time. The aboveground and belowground alfalfa biomass in the T_5_ treatment increased by 30.16% and 34.1%, respectively. The POD, SOD, and CAT activities in the T_5_ treatment of the aboveground alfalfa increased significantly by 73.1%, 77.3%, and 58.8%, respectively. The results indicated that the combination of earthworms and *Bacillus subtilis*-loaded biochar has a strong potential for ameliorating soda saline–alkali soil. The findings of this study demonstrate that the preparation method for *B. subtilis*-loaded biochar is straightforward, yet effective. Both soil incubation experiments and pot trials conducted on soda saline–alkali soil have yielded promising improvement results. Future research should focus on scaling up these experiments to field-level applications. Additionally, this study highlights the need to investigate (1) the synergistic coexistence of *B. subtilis* with other beneficial microorganisms and (2) the potential enhancement of saline–alkali soil remediation efficacy. A promising direction would be to develop biochar composites loaded with multiple beneficial microorganisms, which could potentially achieve more comprehensive and efficient saline–alkali soil rehabilitation.

## Figures and Tables

**Figure 1 microorganisms-13-01243-f001:**
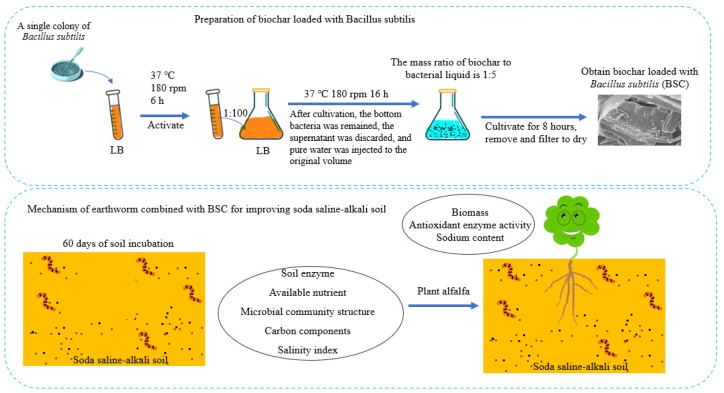
Experimental flowchart of this study.

**Figure 2 microorganisms-13-01243-f002:**
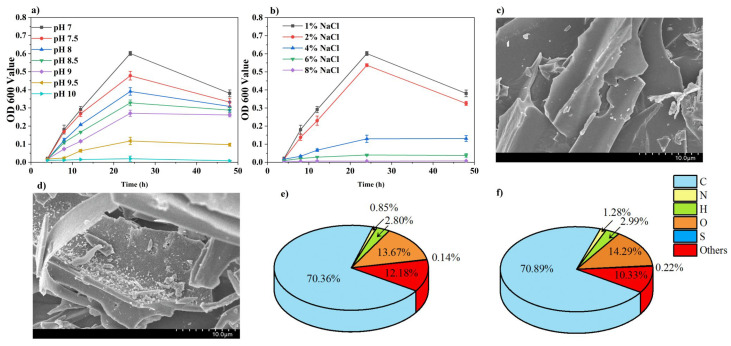
The effects of (**a**) different pH and (**b**) NaCl concentrations on the growth status of free *B. subtilis*, as well as the electron microscopy characterization of (**c**) conventional biochar and (**d**) BSC. Element contents of (**e**) BC and (**f**) BSC.

**Figure 3 microorganisms-13-01243-f003:**
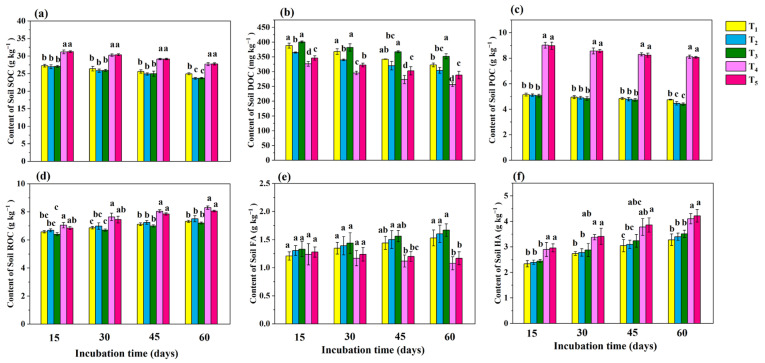
The effects of different treatments on organic carbon components during soil cultivation. (**a**) The content of soil SOC, (**b**) the content of soil DOC, (**c**) the content of soil POC, (**d**) the content of soil ROC, (**e**) the content of soil FA, and (**f**) the content of soil HA. Among them, T_1_ included the addition of 30 g of maize straw and 20 g of cow manure per kilogram of saline–alkali soil; T_2_ included the base of T_1_ and 6 earthworms per kilogram of soil; T_3_ included the base of T_2_ and 25 mL of BS per kilogram of soil; T_4_ included the base of T_2_ and 10 g of conventional biochar per kilogram of soil; and T_5_ included the base of T_2_ and 10 g of BSC per kilogram of soil. Multiple comparisons were performed using Tukey’s HSD test to determine significant differences among groups. Different lowercase letters (e.g., a, b, c, d) indicate statistically significant differences (*p* < 0.05).

**Figure 4 microorganisms-13-01243-f004:**
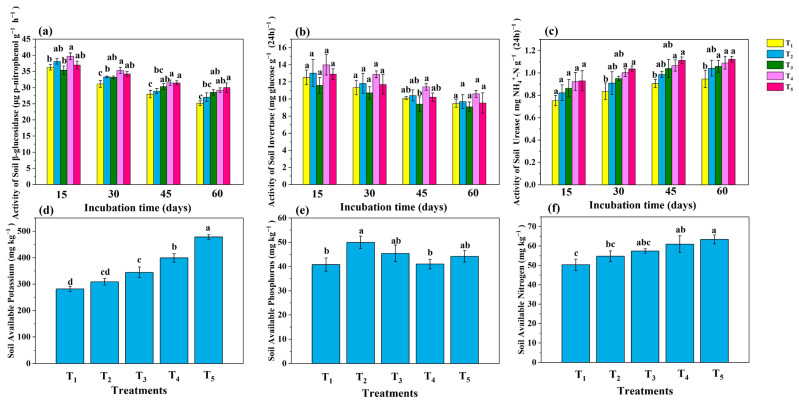
The effects of different treatments during soil cultivation on key soil enzymes and soil available nutrients after 60-day cultivation. (**a**) Soil β-glucosidase activity, (**b**) soil sucrase activity, (**c**) soil urease activity, (**d**) soil available phosphorus content, (**e**) soil available potassium, and (**f**) soil alkaline nitrogen content. Among them, T_1_ included the addition of 30 g of maize straw and 20 g of cow manure per kilogram of saline–alkali soil; T_2_ included the base of T_1_ and 6 earthworms per kilogram of soil; T_3_ included the base of T_2_ and 25 mL of BS per kilogram of soil; T_4_ included the base of T_2_ and 10 g of conventional biochar per kilogram of soil; and T_5_ included the base of T_2_ and 10 g of BSC per kilogram of soil. Multiple comparisons were performed using Tukey’s HSD test to determine significant differences among groups. Different lowercase letters (e.g., a, b, c, d) indicate statistically significant differences (*p* < 0.05).

**Figure 5 microorganisms-13-01243-f005:**
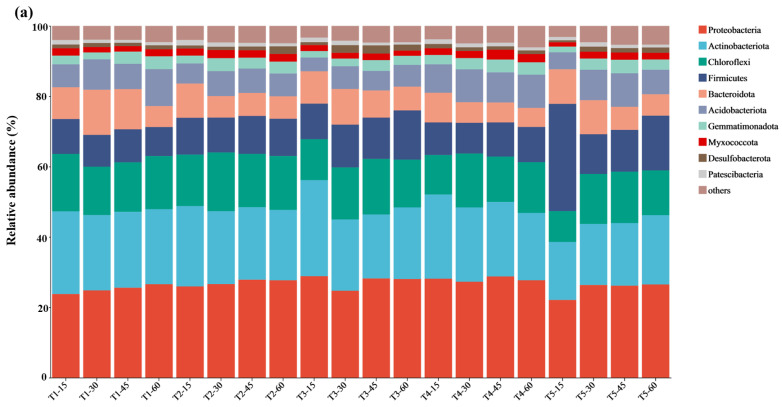
The effect of different treatments on the relative abundance of soil microbial communities during soil cultivation. (**a**) Soil bacterial phylum level, (**b**) soil fungal phylum level, (**c**) soil bacterial genus level, and (**d**) soil fungal genus level. Among them, T_1_ included the addition of 30 g of maize straw and 20 g of cow manure per kilogram of saline–alkali soil; T_2_ included the base of T_1_ and 6 earthworms per kilogram of soil; T_3_ included the base of T_2_ and 25 mL of BS per kilogram of soil; T_4_ included the base of T_2_ and 10 g of conventional biochar per kilogram of soil; and T_5_ included the base of T_2_ and 10 g of BSC per kilogram of soil. The after ‘-number’ represents the time of soil cultivation for that treatment. For example, T_1_-15 represents the relative abundance of microbial communities after 15-day cultivation in T_1_ treatment.

**Figure 6 microorganisms-13-01243-f006:**
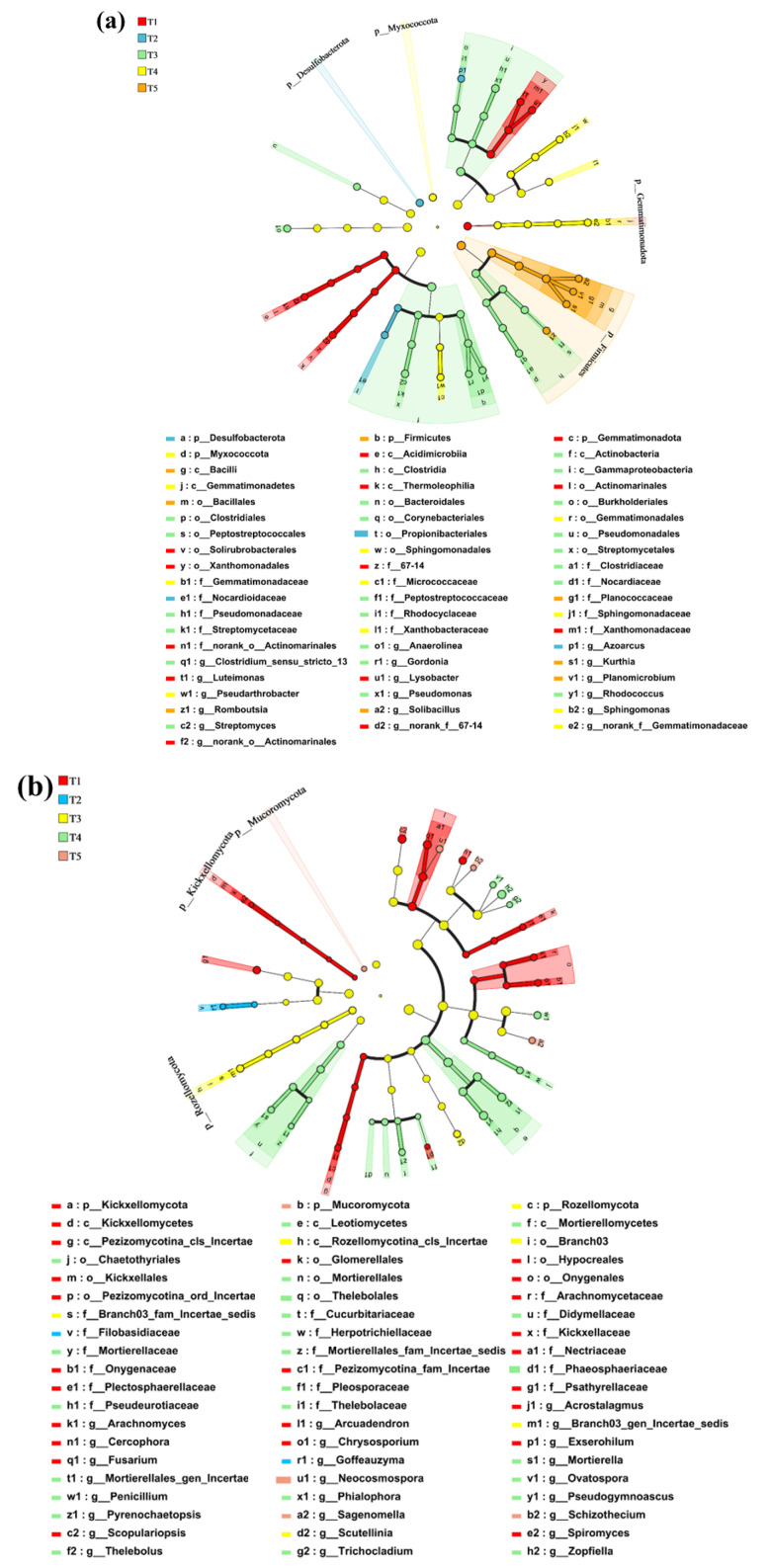
Analysis of soil microbial community differences after 60 days of incubation. (**a**) LefSe analysis of bacterial phylum to genus level differences (LDA > 3.5), (**b**) LefSe analysis of fungal phylum to genus level differences (LDA > 2.0), (**c**) one-way analysis of bacterial genus level differences, (**d**) Venn plot showing differences between bacterial OTUs, (**e**) one-way analysis of fungal genus level differences, (**f**) Venn plot showing differences between fungal OTUs. Among them, T_1_ included the addition of 30 g of maize straw and 20 g of cow manure per kilogram of saline–alkali soil; T_2_ included the base of T_1_ and 6 earthworms per kilogram of soil; T_3_ included the base of T_2_ and 25 mL of BS per kilogram of soil; T_4_ included the base of T_2_ and 10 g of conventional biochar per kilogram of soil; and T_5_ included the base of T_2_ and 10 g of BSC per kilogram of soil. The symbols *, ** and *** present differences at levels 0.05, 0.01 and 0.001, respectively.

**Figure 7 microorganisms-13-01243-f007:**
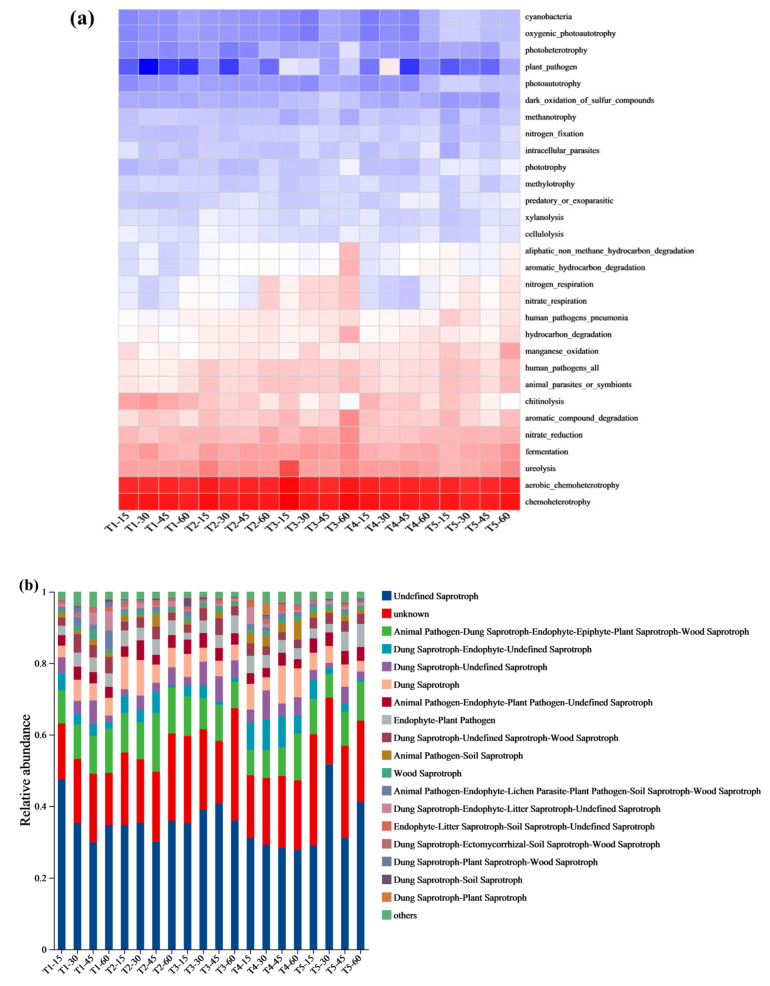
Prediction and correlation of soil microbial function, (**a**) bacterial FAPROTAX prediction, (**b**) fungal FUNGuild prediction, and (**c**) correlation between soil nutrients and salinity indicators and microorganisms (taken from microbial samples on day 60). Among them, T_1_ included the addition of 30 g of maize straw and 20 g of cow manure per kilogram of saline–alkali soil; T_2_ included the base of T_1_ and 6 earthworms per kilogram of soil; T_3_ included the base of T_2_ and 25 mL of BS per kilogram of soil; T_4_ included the base of T_2_ and 10 g of conventional biochar per kilogram of soil; and T_5_ included the base of T_2_ and 10 g of BSC per kilogram of soil. The after ‘-number’ represents the time of soil cultivation for that treatment. For example, T_1_-15 represents the relative abundance of microbial communities after 15 days of cultivation in the T_1_ treatment.

**Figure 8 microorganisms-13-01243-f008:**
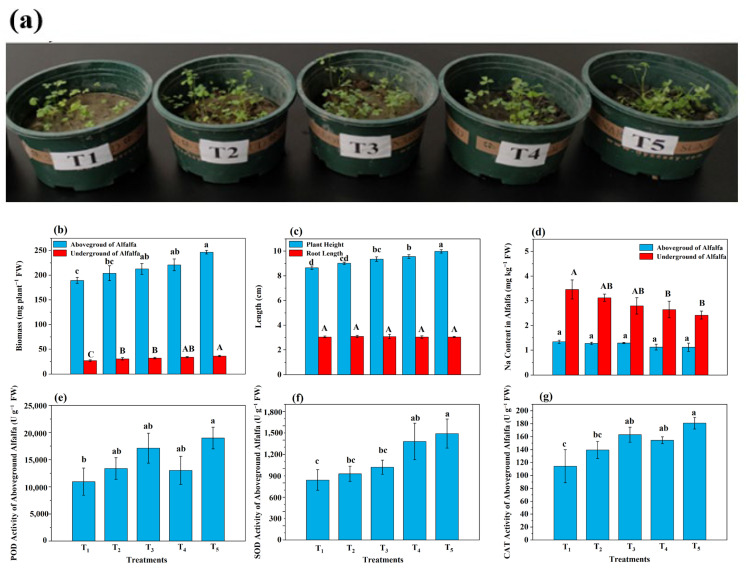
The effects of different treatments on the physical and chemical indicators of alfalfa growth. (**a**) Alfalfa growth status, (**b**) alfalfa biomass, (**c**) alfalfa plant height and root length, (**d**) alfalfa Na content, (**e**) alfalfa above-ground POD activity, (**f**) alfalfa above-ground SOD activity, and (**g**) alfalfa above-ground CAT activity. Among them, T_1_ included the addition of 30 g of maize straw and 20 g of cow manure per kilogram of saline–alkali soil; T_2_ included the base of T_1_ and 6 earthworms per kilogram of soil; T_3_ included the base of T_2_ and 25 mL of BS per kilogram of soil; T_4_ included the base of T_2_ and 10 g of conventional biochar per kilogram of soil; and T_5_ included the base of T_2_ and 10 g of BSC per kilogram of soil. Multiple comparisons were performed using Tukey’s HSD test to determine significant differences among groups. Different lowercase letters (e.g., a, b, c, d) indicate statistically significant differences (*p* < 0.05) for the aboveground part of alfalfa, and capital letter (e.g., A, B, C) indicate statistically significant differences (*p* < 0.05) for the underground part of alfalfa.

**Figure 9 microorganisms-13-01243-f009:**
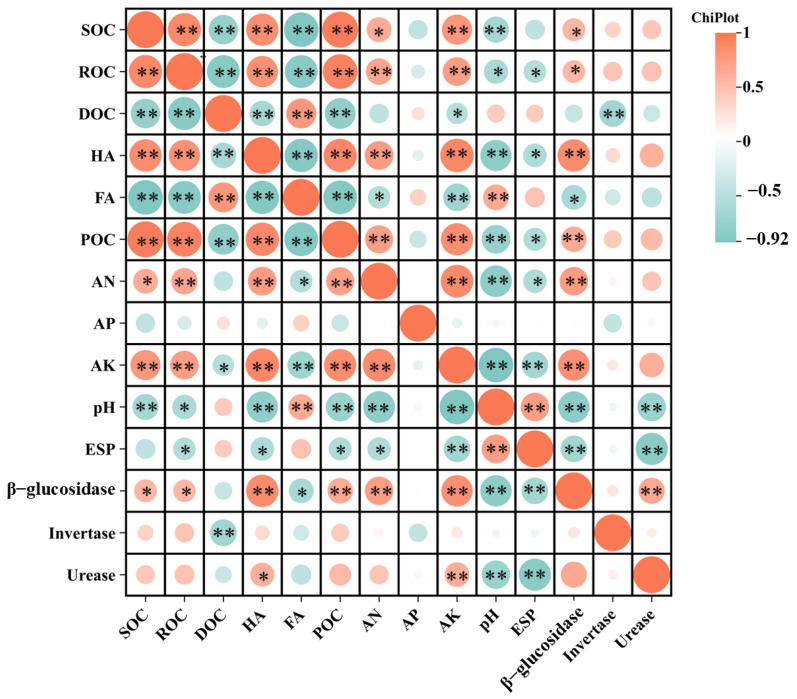
Correlation analysis between soil indexes. Note: The symbols * and ** present differences at levels 0.05 and 0.01, respectively. The data in the figure were obtained after 60 days of cultivation.

**Table 1 microorganisms-13-01243-t001:** Elimination of salt alkali barriers by different treatments.

Treatments	pH	EC (μs cm^−1^)	$({\mathrm{CO}}_{3}^{2-}+{\mathrm{HCO}}_{3}^{-}$) (mg kg^−1^)	ENa (mmol kg^−1^)	CEC (mmol kg^−1^)	ESP (%)
T_1_	9.32 ± 0.04 a	651 ± 12.1 b	1088 ± 13.4 a	2.43 ± 0.08 a	18.7 ± 1.93 c	13.1 ± 1.66 a
T_2_	9.27 ± 0.03 ab	466 ± 22.0 d	1042 ± 17.6 b	2.18 ± 0.06 c	19.2 ± 2.01 c	11.6 ± 1.55 ab
T_3_	9.20 ± 0.03 bc	424 ± 15.6 d	1003 ± 17.0 bc	2.20 ± 0.07 c	20.6 ± 1.28 ab	10.6 ± 0.43 ab
T_4_	9.17 ± 0.03 c	824 ± 26.1 a	980 ± 11.0 c	2.30 ± 0.04 ab	23.0 ± 1.39 ab	10.0 ± 0.43 b
T_5_	9.08 ± 0.03 d	539 ± 15.1 c	968 ± 13.0 c	2.32 ± 0.07 ab	24.1 ± 2.15 a	9.77 ± 0.88 b

Among them, T_1_ included the addition of 30 g of maize straw and 20 g of cow manure per kilogram of saline–alkali soil; T_2_ included the base of T_1_ and 6 earthworms per kilogram of soil; T_3_ included the base of T_2_ and 25 mL of BS per kilogram of soil; T_4_ included the base of T_2_ and 10 g of conventional biochar per kilogram of soil; and T_5_ included the base of T_2_ and 10 g of BSC per kilogram of soil. Multiple comparisons were performed using Tukey’s HSD test to determine significant differences among groups. Different lowercase letters (e.g., a, b, c, d) indicate statistically significant differences (*p* < 0.05).

**Table 2 microorganisms-13-01243-t002:** Microbial alpha diversity under different groups.

Groups	Bacterial			Fungal		
	Ace	Chao1	Shannon	Ace	Chao1	Shannon
T_1_-15	4783 ± 107 a	4602 ± 111 a	6.84 ± 0.08 a	552 ± 54.8 a	550 ± 42.5 a	3.48 ± 0.46 a
T_2_-15	4819 ± 149 a	4600 ± 160 a	6.82 ± 0.05 a	591 ± 39.0 a	592 ± 33.3 a	3.71 ± 0.14 a
T_3_-15	4413 ± 209 a	4224 ± 186 ab	6.47 ± 0.34 ab	551 ± 28.3 a	535 ± 25.3 a	3.71 ± 0.11 a
T_4_-15	4818 ± 183 a	4596 ± 197 a	6.86 ± 0.03 a	514 ± 17.3 a	502 ± 20.3 a	3.75 ± 0.10 a
T_5_-15	4071 ± 231 a	3843 ± 224 b	5.66 ± 0.22 b	527 ± 8.38 a	527 ± 10.3 a	3.75 ± 0.31 a
T_1_-30	4584 ± 178 a	4408 ± 229 a	6.60 ± 0.09 b	518 ± 64.2 a	518 ± 68.3 a	3.80 ± 0.16 a
T_2_-30	4762 ± 235 a	4574 ± 193 a	6.91 ± 0.10 a	488 ± 41.1 a	485 ± 38.3 a	3.63 ± 0.39 a
T_3_-30	4614 ± 100 a	4453 ± 73.0 a	6.75 ± 0.07 ab	480 ± 37.7 a	480 ± 35.6 a	3.38 ± 0.23 a
T_4_-30	5024 ± 264 a	4835 ± 305 a	6.90 ± 0.13 a	537 ± 25.3 a	545 ± 20.3 a	3.72 ± 0.11 a
T_5_-30	4584 ± 64.5 a	4403 ± 79.8 a	6.78 ± 0.05 ab	493 ± 19.4 a	494 ± 21.0 a	3.51 ± 0.40 a
T_1_-45	4743 ± 224 b	4557 ± 172 b	6.77 ± 0.09 b	512 ± 56.6 a	518 ± 60.8 a	3.88 ± 0.15 a
T_2_-45	4891 ± 73.5 ab	4707 ± 129 ab	6.86 ± 0.05 ab	487 ± 37.7 a	497 ± 41.8 a	3.70 ± 0.18 a
T_3_-45	4727 ± 75.3 b	4549 ± 136 b	6.81 ± 0.05 b	464 ± 46.1 a	461 ± 48.2 a	3.52 ± 0.32 a
T_4_-45	5101 ± 266 a	4899 ± 240 a	6.94 ± 0.03 a	509 ± 36.0 a	512 ± 36.3 a	3.83 ± 0.26 a
T_5_-45	4990 ± 106 ab	4810 ± 119 ab	6.86 ± 0.03 ab	526 ± 47.8 a	529 ± 46.7 a	3.79 ± 0.21 a
T_1_-60	4969 ± 208 a	4796 ± 185 a	6.85 ± 0.04 abc	454 ± 65.9 a	449 ± 67.4 a	3.58 ± 0.30 a
T_2_-60	5163 ± 81.3 a	4986 ± 20.4 a	6.96 ± 0.04 ab	460 ± 59.1 a	461 ± 49.9 a	3.58 ± 0.40 a
T_3_-60	4803 ± 196 a	4623 ± 185 a	6.72 ± 0.16 c	441 ± 13.0 a	445 ± 18.8 a	3.57 ± 0.23 a
T_4_-60	5224 ± 146 a	5025 ± 181 a	7.01 ± 0.04 a	510 ± 33.0 a	509 ± 35.8 a	3.80 ± 0.15 a
T_5_-60	4828 ± 185 a	4611 ± 139 a	6.77 ± 0.08 bc	518 ± 69.4 a	516 ± 56.5 a	3.54 ± 0.24 a

Among them, T_1_ included the addition of 30 g of maize straw and 20 g of cow manure per kilogram of saline–alkali soil; T_2_ included the base of T_1_ and 6 earthworms per kilogram of soil; T_3_ included the base of T_2_ and 25 mL of BS per kilogram of soil; T_4_ included the base of T_2_ and 10 g of conventional biochar per kilogram of soil; and T_5_ included the base of T_2_ and 10 g of BSC per kilogram of soil. Multiple comparisons were performed using Tukey’s HSD test to determine significant differences among groups. Different lowercase letters (e.g., a, b, c) indicate statistically significant differences (*p* < 0.05).

## Data Availability

The original contributions presented in this study are included in the article/Appendix A. Further inquiries can be directed to the corresponding author.

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
