# Peer review of "The Improved Remediation Effect of the Combined Use of Earthworms with Bacillus subtilis-Loaded Biochar in Ameliorating Soda Saline–Alkali Soil"

_microorganisms, 2025, doi:10.3390/microorganisms13061243_

Round 1

Reviewer 1 Report

Comments and Suggestions for Authors

The work concerns the important problem of using sodic-salty-alkaline soil. The work is well planned and contains a large number of results. The work is interesting and has practical applications. The quality of the work is good, but the reviewer indicates elements that should be improved:

  • 1 Materials – Line 120 „soil used in this study was collected from Changling County, Jilin Province, China” - Please write the geographic coordinates of where exactly the samples were taken from. How was a representative sample obtained? Describe the methodology and procedure for soil collection and storage
  • Lines 125-126 – „Air-dried cow manure and maize straw were collected from local farms, and their physicochemical properties are described by Chen et al. [33]” - Please describe how to obtain a representative sample of manure and the basic properties of manure. Of course, you can refer to the details described in the article by Chen et al., but it is worth adding the basic information in the present article.
  • 6 Statistical analysis- Why do the authors use Duncan’s multiple range test for statistical analysis?
  • Materials and methods - Due to the large number of studies conducted and the large number of procedures used, the reviewer believes that it would be good to include a flowchart presenting the overall experimental procedure in the work in the form of graphics or block diagrams
  • Figures should be placed after the text in which it is mentioned, not before it.
  • 5 c-f please increase the font size of the text and numbers in the chart
  • Discussion- I ask the authors to consider including a table at the end of the discussion that would synthetically show which systems tested had the best potential to improve the quality of sodic-salt-alkaline soil.
  • Conclusions - How can the results be used practically? What are the future research perspectives? Should the experiment be carried out in real conditions in the agricultural field? Should other microorganisms or fertilizers be used?

Author Response

I have written the responses to the comments of the two reviewers in one Word and uploaded it.

Reviewer 2 Report

Comments and Suggestions for Authors

Combined use of earthworm and Bacillus loaded biochar is a good idea for amelioration of soda saline alkali soil. I think this paper is worth to be published in this Journal. But as for statistical analysis, Duncan's MRT is now not recommended in the standpoint of safety. I recommend using Tuley or other modern analysis to elucidate merit of the results. I added my comments on the paper directly. 

Author Response

(The authors gave the same response as above.)
